# Dissecting Larval Zebrafish Hunting using Deep Reinforcement Learning Trained RNN Agents

## Abstract

Larval zebrafish hunting provides a tractable setting to study how ecological and energetic constraints shape adaptive behavior in both biological brains and artificial agents. Here we develop a minimal agent-based model, training recurrent policies with deep reinforcement learning in a bout-based zebrafish simulator. Despite its simplicity, the model reproduces hallmark hunting behaviors—including eye vergence-linked pursuit, speed modulation, and stereotyped approach trajectories—that closely match real larval zebrafish. Quantitative trajectory analyses show that pursuit bouts systematically reduce prey angle by roughly half before strike, consistent with measurements. Virtual experiments and parameter sweeps vary ecological and energetic constraints, bout kinematics (coupled vs. uncoupled turns and forward motion), and environmental factors such as food density, food speed, and vergence limits. These manipulations reveal how constraints and environments shape pursuit dynamics, strike success, and abort rates, yielding falsifiable predictions for neuroscience experiments. These sweeps identify a compact set of constraints—binocular sensing, the coupling of forward speed and turning in bout kinematics, and modest energetic costs on locomotion and vergence—that are sufficient for zebrafish-like hunting to emerge. Strikingly, these behaviors arise in minimal agents without detailed biomechanics, fluid dynamics, circuit realism, or imitation learning from real zebrafish data. Taken together, this work provides a normative account of zebrafish hunting as the optimal balance between energetic cost and sensory benefit, highlighting the trade-offs that structure vergence and trajectory dynamics. We establish a *virtual lab* that narrows the experimental search space and generates falsifiable predictions about behavior and neural coding.

## 1 Introduction

Adaptive behavior unfolds under ecological and energetic constraints that shape what strategies are feasible or optimal (Zhu & Goodhill, 2023; Bolton et al., 2019). Understanding how such constraints give rise to structured behavioral sequences is a shared challenge for neuroscience, neuroAI, and artificial intelligence (Singh, 2021; Huang et al.).

Larval zebrafish hunting is a particularly clear example of structured behavior: animals pursue prey through discrete bouts organized into exploration, orientation, pursuit, and either strike or abort (Bianco et al., 2011; Johnson et al., 2020; Bolton et al., 2019). These behaviors exhibit consistent hallmarks, including a vergence-linked shift into a "hunting mode" (Bianco et al., 2011; Johnson et al., 2020), systematic halving of prey angle across pursuit bouts (Bolton et al., 2019), and stereotyped approach trajectories (Johnson et al., 2020). Despite this well-documented structure, it remains unclear why these behavioral motifs emerge and persist, or under what constraints they represent optimal solutions (Zhu & Goodhill, 2023). In particular, we lack explanations for why vergence angles shift abruptly (Johnson et al., 2020), why prey angle is consistently reduced by about 50% per pursuit bout (Bolton et al., 2019), and why some hunts succeed while others abort (Zhu & Goodhill, 2023). Prior computational accounts, including bounded integrator models (Bahl & Engert, 2020) and probabilistic inference frameworks (Bolton et al., 2019), capture aspects of zebrafish hunting but stop short of explaining why stereotyped trajectories emerge as optimal strategies.

Models of behavior in neuroscience are often *descriptive*, specifying what actions animals perform under given conditions (Johnson et al., 2020, e.g.,), or *mechanistic*, capturing how neural circuits generate those behaviors (Rajan et al., 2016, e.g.,). In contrast, our approach is *normative*: it explains *why* a behavioral strategy emerges as optimal given specified constraints, in line with reinforcement-learning–based normative models of perception and decision-making (Bahl & Engert, 2020; Haesemeyer et al., 2019, e.g.,).

Although larval zebrafish are unusually accessible for circuit-level and behavioral experiments thanks to their transparency and genetic tools (Randlett et al., 2015; Leyden et al., 2021), careful ethological work still leaves key variables hard to isolate and manipulate (Johnson et al., 2020; Zhu & Goodhill, 2023). Virtual-reality assays can reliably evoke components, such as convergent eye movements and orienting turns, but typically do not permit fine-grained, closed-loop control over the entire hunting sequence (Bianco et al., 2011). In naturalistic arenas, ecological variables such as prey density, prey kinematics, and energetic costs covary, making it difficult to vary them systematically one at a time for causal inference. Moreover, internal state variables—such as motivational drive or accumulated evidence—that likely govern the transition between pursuit, strike, and abort remain difficult to access with current experimental methods (Bahl & Engert, 2020; Randlett et al., 2019).

Task-optimized artificial neural networks provide a complementary approach, offering a way to test how specific constraints give rise to structured behavior when direct experimentation falls short (Haesemeyer et al., 2019; Singh, 2021). Prior studies show that recurrent neural network (RNN) agents trained with deep reinforcement learning (DRL) can capture biological strategies, such as electrosensory navigation in weakly electric fish (Johnson-Yu et al., 2024). The same methods have also been applied in AI and NeuroAI settings, where they give rise to complex planning behaviors and structured internal representations (Huang et al.; 2025; Simmons-Edler et al., 2025; Singh et al., 2023; Keller et al., 2025). Taken together, this body of work motivates applying task-optimized recurrent agents to zebrafish hunting as a principled way to probe how ecological and energetic constraints shape structured behavior.

Here, we introduce a biologically inspired hunting simulator where RNN-based DRL agents learn to pursue prey through discrete bouts (with prey modeled as stochastic walkers mimicking paramecia rather than adversarial agents (Zocchi et al., 2025)). This framework enables systematic manipulations of ecological variables, sensory constraints, and energetic costs, providing a virtual laboratory for uncovering how constraints yield structured behavior. The same approach—task-optimized DRL agents analyzed via structured sweeps—can extend beyond zebrafish hunting to other sensorimotor systems and inform inductive biases in AI and robotics.

**Our key contributions are:**
(i) We introduce a biologically inspired *framework* in which virtual zebrafish agents perceive, move, and hunt through discrete bouts, enabling systematic manipulation of ecological, sensory, and energetic constraints.
(ii) We train recurrent agents with deep reinforcement learning and show that they spontaneously develop naturalistic hunting strategies without imitation learning from real zebrafish data.
(iii) Through detailed behavioral analyses and parameter sweeps, we identify a compact set of constraints—binocular sensing, bout kinematics, and energetic costs—minimal yet sufficient for zebrafish-like hunting behavior to emerge.
(iv) We establish a *virtual lab* that provides falsifiable predictions for in vivo experiments, offers a normative account of why hunting stereotypy emerges, and serves as a starting point for probing how task-trained RNNs might encode behavioral state variables.

## 2 METHODS

### 2.1 BIOLOGICALLY INSPIRED ENVIRONMENT AND AGENT MODEL

Following experimental studies of larval zebrafish in open circular arenas (Bahl & Engert, 2020; Harpaz et al., 2021), we simulate a two-dimensional circular arena with rigid boundaries and a diameter uniformly sampled from 33–100 mm. The arena contains a fixed number of stochastically moving prey agents, with both the prey and the zebrafish agent initialized uniformly at random within the arena (Fig. 1a, left). Prey motion statistics are chosen to approximate *Paramecium* swimming behavior observed in feeding experiments (Johnson et al., 2020). The agent (Fig 1a, right) consists of

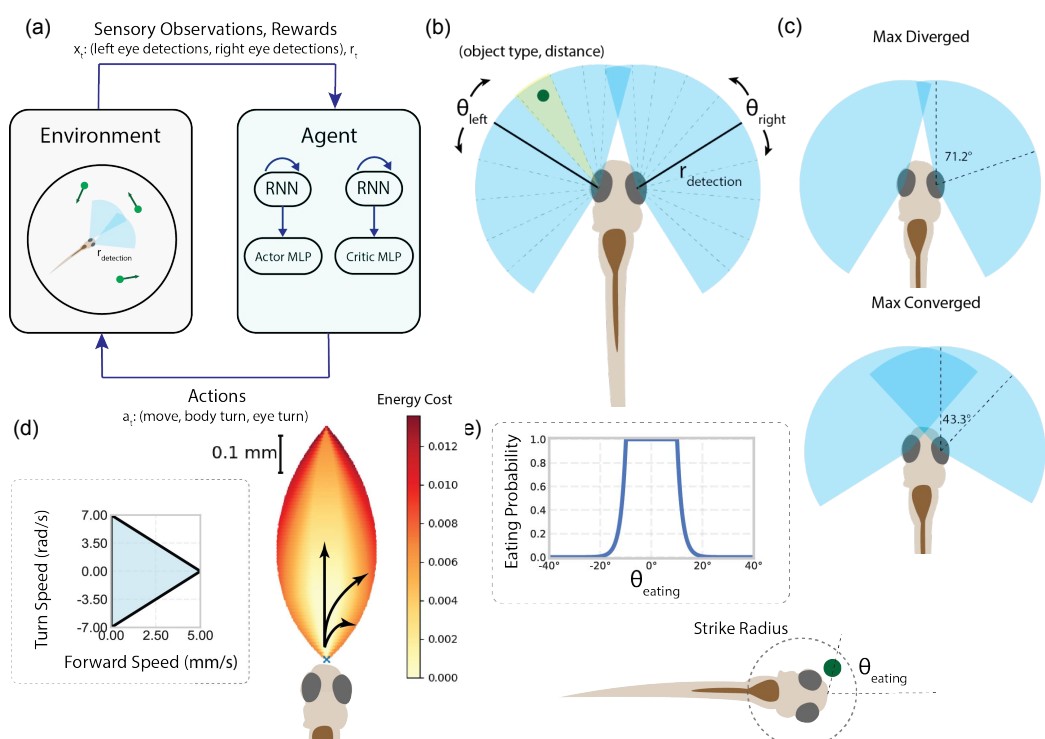

Figure 1: *A biologically inspired DRL framework grounds recurrent agent perception, actions, and rewards in zebrafish hunting.* (a) Closed-loop setup: an actor–critic RNN policy selects actions (forward speed, turn speed, vergence angle) that update the environment, which in turn provides new observations and rewards. (b) Zebrafish visual model: each eye has a 163° monocular field of view, subdivided into 10 angular sectors that report the nearest object type and distance. (c) Eye vergence: eyes rotate between empirically observed divergence and convergence limits, defining the binocular overlap region. (d) Triangular action space: linear (forward) and angular (turn) speeds are coupled; larger forward speeds permit only smaller turns. A linear energy cost is imposed upon movements surpassing a fixed turn or forward speed. (e) Strike model: when prey are within strike radius, capture probability decays with alignment error $|\theta|$ according to a modified Laplace distribution fit to empirical strike angle data.

two recurrent neural networks (RNN) of 256 units each (Rajan et al., 2016) and parallel two-layer Actor and Critic Multi-Layer Perceptrons (MLPs) (Ni et al., 2021). The agent acts in closed-loop with the environment, generating actions that change the environment state, which in turn provides sensory observations and rewards.

We model the agent with two eyes with a fixed forward offset and width, with each eye having a monocular field of view of 163 degrees and a detection radius of 10 mm matched to empirical larval zebrafish visual field experiments (Bianco et al., 2011). Each eye's field of view is divided into 10 angular sectors. Sector width increases with radial distance to model reduced spatial resolution (Figure 1b). Each sector receives object type and distance information to the closest object (food, wall) in that sector. The agent can rotate its eyes to change the vergence angle, which defines the size of the binocular region. The maximum and minimum vergence values are set according to experimental data (Fig. 1c). We use coupled eyes which share a single vergence control $\theta_v \in [\theta_{\min}, \theta_{\max}]$, with mirrored symmetry about the midline: $\theta_L = -\theta_v, \quad \theta_R = \theta_v$.

At each bout the agent receives an observation vector consisting of 10 inputs per eye sector: object type (prey/wall/none) and normalized distance in [0,1]. If no object is detected the sector is zeroed. Binocular sectors contribute two independent samples. At each bout, the reported distance in a sector is perturbed multiplicatively, $\hat{d} = d(1 + \epsilon)$ with $\epsilon \sim \text{Uniform}[-\sigma_d, \sigma_d]$, where $\sigma_d$ is the distance noise parameter (Appendix 6.4). The angle (and hence the sector that fires) perceived to the

object is also noisy, $\hat{\theta} = \theta + \epsilon$ with $\epsilon \sim \mathrm{Uniform}[-\sigma_\theta, \sigma_\theta]$, where $\sigma_\theta$ is the angle noise parameter (Appendix 6.4). Because binocular sectors yield two independent samples (one from each eye), the binocular region is less noisy than the monocular regions. Additionally, distance estimates are supplied only in the binocular field, while monocular sectors provide binary detection (present/absent) without distance. We also allow angular noise in monocular sensing. Proprioceptive state includes current vergence angles and previous bout action. The final input dimensionality is 45. These per-sector measurements, together with proprioceptive state, are concatenated into an observation vector fed to the network.

We use a discrete time step equal to the duration of one bout ($\Delta t = 125$ ms), following (Johnson et al., 2020). Linear speed and angular speed per bout are coupled. Larger forward speeds permit only smaller turns, creating a triangular action space (Fig. 1d). When within strike radius, capture probability decays with absolute alignment error $|\theta|$ according to a modified Laplace form (Figure 1e), with the decay parameter chosen to match the empirical distribution of prey angles at strike in larvae (Bolton et al., 2019). Thus, the policy outputs are forward speed, turn speed, and vergence angle, subject to constraints below.

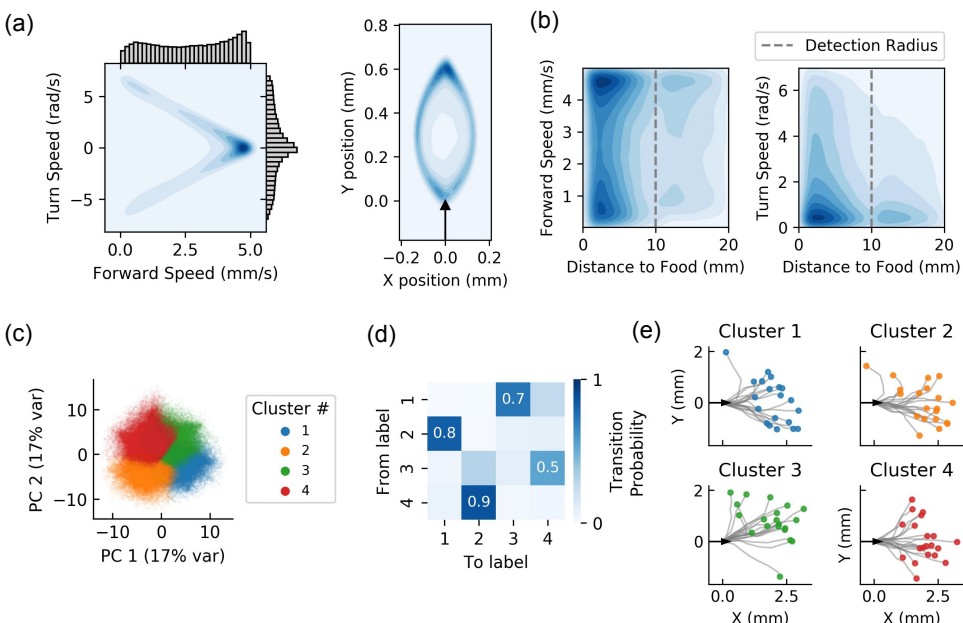

Figure 2: *Agent movement statistics reveal distinct bout types resembling real zebrafish strikes and adjustments.* **(a)** Agent action selection: density plot and histograms of forward and turn speeds chosen across evaluation (left) along with spatial displacement after one bout (right). **(b)** Forward and turn speeds as a function of distance to prey: two distinct clusters appear when prey are nearby—fast, straight bouts (strikes) and slower, variable-turn bouts (fine adjustments). **(c)** Behavioral motifs across timesteps: Principal component analysis (PCA) of movement trajectories (8 timesteps, 1 s) clustered into five groups with K-means (PC1: 17% variance, PC2: 17%); see Appendix 6.3. **(d)** Transition probabilities between clusters: the transition matrix is characterized by a dominant outgoing probability at each state, suggesting that bout sequences are stereotyped. **(e)** Example trajectories: 20 example trajectories (1 s each) from the five clusters. Together these analyses show that the agent's bout repertoire is structured into discrete modes that align with known zebrafish hunting motifs.

## 2.2 CONSTRAINTS AND REWARDS THAT INCENTIVIZE NATURALISTIC HUNTING

We develop a framework that parameterizes key ecological and sensory constraints to determine which features make naturalistic zebrafish-like hunting the optimal strategy under our objective and environment statistics. This provides a normative account of why larval zebrafish exhibit a distinctly stereotyped hunting mode.

**Speed cost:** Forward speeds above a fixed threshold $v_{\text{th}}$ incur an energetic penalty, $C_{\text{speed}} = \beta_{\text{speed}} \max(0, v_t - v_{\text{th}})$, where $v_t$ is the chosen forward speed during a bout and $\beta_{\text{speed}}$ sets the penalty magnitude (Appendix 6.4). This models the energetic cost and physiological limitation of sustained high-speed swimming in larval zebrafish.

**Turn cost:** Angular speeds above a fixed threshold $\omega_{\text{th}}$ incur an energetic penalty, $C_{\text{turn}} = \beta_{\text{turn}} \max(0, |\omega_t| - \omega_{\text{th}})$, where $\omega_t$ is the chosen angular speed and $\beta_{\text{turn}}$ sets the penalty magnitude (Appendix 6.4). This mirrors the speed cost term but applies to turning.

**Vergence cost:** Eye positions deviating from their resting (divergent) angles incur an energetic penalty, $C_{\text{vergence}} = \alpha_{\text{eye}}\left(|\theta_L - \theta_{L,\text{rest}}| + |\theta_R - \theta_{R,\text{rest}}|\right)$, where $\theta_L$ and $\theta_R$ are the chosen vergence angles for left and right eyes, $\theta_{L,\text{rest}}$ and $\theta_{R,\text{rest}}$ are their respective resting angles, and $\alpha_{\text{eye}}$ scales the penalty magnitude (Appendix 6.4). This term models the muscular effort required to maintain converged eye positions in larval zebrafish.

Thus, the per-timestep reward is $R_t = R_{\text{capture}} - C_{speed} - C_{turn} - C_{vergence} + R_{shape}$ The primary reward is obtained on successful prey capture. We also provide a weak distance-based shaping reward $R_{shape}$ that helps with convergence. All parameter values are listed in Appendix 6.4.

## 2.3 AGENT TRAINING

We train with Proximal Policy Optimization (PPO) (Schulman et al., 2017) on 4M simulation steps using a linear curriculum where prey density decreased, prey motion variability increased, and strike tolerance narrowed with training progress. Baseline hyperparameters were identified through exploratory tuning and later tested for sensitivity in Section 3.5. To account for stochasticity in training, we initialized 10 independent runs with different random seeds under the baseline setting and, analogous to evolutionary selection for fitness, carried forward the best-performing seed for detailed behavioral analyses in Sections 3.1–3.4.

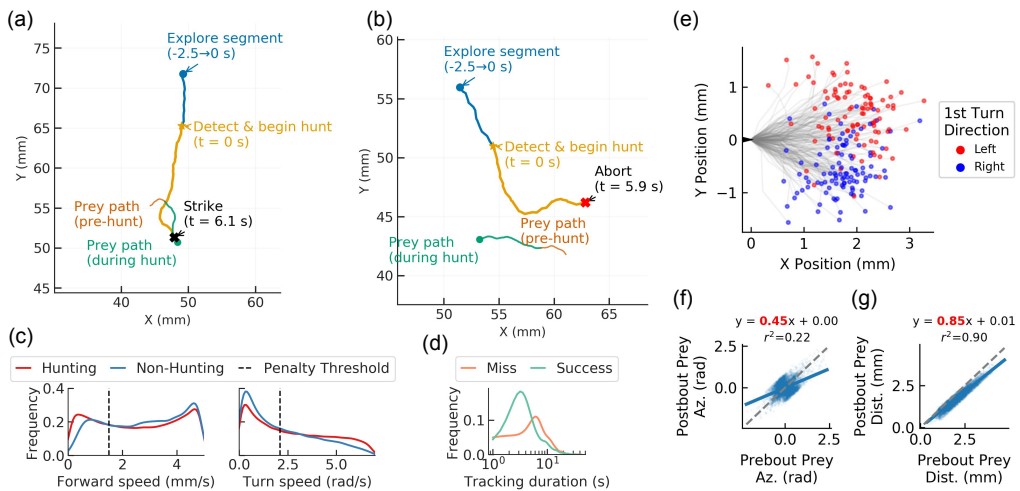

Figure 3: *Successful and failed hunts diverge in bout sequences, durations, and pursuit corrections.* **(a)** Example trajectory ending in a successful strike. **(b)** Example trajectory ending in an abort. **(c)** Distribution of forward and turn speeds during hunting versus exploration, showing larger turn speeds and more precise forward bouts during hunts. **(d)** Hunt durations separated by successful versus failed hunts: successful hunts are typically shorter, with aborts occurring after extended tracking durations without successful capture. **(e)** 200 sample trajectories in the 0.75 s (6 bouts) preceding prey capture, colored by initial turn direction. **(f–g)** Bout-wise changes in prey azimuth and distance across the final 0.75 s of successful hunts: each bout reduces angle to prey by about half and prey distance by about 15%, closely matching empirical zebrafish hunting data (Bolton et al., 2019). Together, these demonstrate that successful hunts are characterized by pursuit with systematic angle reduction.

## 3 RESULTS

### 3.1 AGENT MOVEMENTS RESEMBLE REAL LARVAL ZEBRAFISH BOUTS

For our framework to have explanatory value, a key test is whether agents trained primarily for prey capture, under movement constraints, develop bout repertoires that look like those of real zebrafish. As seen in Fig. 2, we see that trained agents indeed develop a structured bout repertoire that mirrors the discrete movement patterns observed in larval zebrafish. Agents show rich variability in individual bouts (Fig. 2a) that is modulated by sensory context (Fig. 2b). Specifically, as prey appear within detection range, we see a high variability in movement speed, corresponding to two types of bouts: (i) fast, nearly straight bouts that correspond to strikes, and (ii) slower bouts with variable turning, which serve to refine alignment to prey. Over sequences of bouts, the joint distribution of forward and turn sequences organizes into distinct clusters corresponding to straight swimming, left and right reorientations, and intermediate adjustments (Fig. 2c-e). Unsupervised clustering of short trajectories confirms this structure (Fig. 2c-d) (See Appendix 6.3 for details). We find that agent execute stereotyped sequences of movements as they explore an hunt (Fig. 2d), just as with real larval zebrafish (Mearns et al., 2020; Marques et al., 2018). Our analyses demonstrate that the agent's movement statistics self-organize into a rich repertoire that closely resembles real zebrafish bout classes.

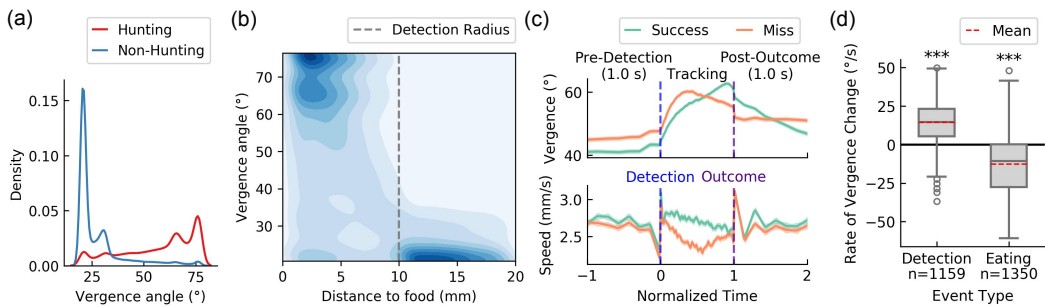

Figure 4: ***Vergence dynamics reveal costly but advantageous binocular sensing during hunts.***
**(a)** Distribution of vergence angles during successful hunts versus non-hunting periods: hunts show consistently higher vergence. **(b)** Distribution of vergence angles as a function of distance to food: agents exhibit high eye vergence angles when food is within their detection range and low vergence when it is not. **(c)** Time course of average vergence angle over all episodes aligned to prey detection and strike/abort: vergence rises sharply at hunt onset, peaks before strike, and relaxes afterward. Error bars represent 1 standard error. Time corresponds to 1 s before detection and after outcome, and all tracking sequences in between normalized to equal duration. $n = 1366$ successful hunts; $n = 1554$ misses. **(d)** Rate of vergence change within 1 s after prey detection and successful eating events: Vergence increases at an average rate of $14.5°$/s in the 1 s after detection (Student's $t$-test, $n = 1159$ detection events) and decreases at an average rate of $-12.7°$/s following eating ($n = 1350$ eating events). Together these results show that although convergence incurs a metabolic cost, agents adopt and sustain it during hunts because binocular sensing improves prey localization, paralleling empirical zebrafish data (Bianco et al., 2011; Johnson et al., 2020).

### 3.2 MOVEMENT STATISTICS VARY ACROSS SUCCESSFUL HUNTS, FAILED HUNTS, AND EXPLORATION

A hunting sequence begins when a prey item is first detected and ends either with a capture (strike) or when the prey exits the perception radius without capture (abort). All other periods are defined as exploration. Fig. 3a and 3b show sample hunting trajectories that end in a successful strike and an abort, respectively. Fig. 3c depicts the agent's forward speed and turn speed distributions when hunting vs. not hunting, evaluated across a set of 200 fixed arena sizes, food locations, and agent initializations. During hunting, the agent more frequently executes high speed turns at a higher frequency and large forward bouts less often, indicating a pursuit strategy that prioritizes proper orientation over speed. Alternatively, when exploring, the agent tends to prioritize forward speed

over turn speed. We see that successful strikes have a shorter tracking duration on average than unsuccessful strikes (Fig. 3d). This suggests that the agent aborts pursuits after chasing prey for extended periods of time, choosing instead to explore for other targets. Agent movement immediately before successful strikes mimics the movement of real larval zebrafish when hunting. As agents pursue their prey, they tend to keep their target on the same side of their field of view, seldom oversteering (Fig. 3e). With each successive bout, the agent's average distance to the prey is reduced by about 15% and its angular alignment error is approximately halved (Fig. 3f-g). This recursive hunting pattern closely matches the pursuit strategy of larval zebrafish (Bolton et al., 2019).

### 3.3 Eye Convergence During Hunting Mirrors Empirical Zebrafish Data

Agents show higher eye vergence during successful hunts compared to non-hunting periods (Fig. 4a). Furthermore, differences in vergence behavior can be seen as a function of agent distance to food (Fig. 4b), with agents adopting high vergence when food is within the detection radius and low vergence when food is outside. During hunting, vergence angle increases steadily and peaks just before the strike (Fig. 4c), closely matching empirical zebrafish data (Bianco et al., 2011). This suggests that the agent accepts the energetic cost of deviating from the resting eye position in exchange for improved sensory information. Before prey detection, vergence angles are similar in successful and failed hunts. Midway through tracking, the trajectories diverge: successful hunts continue to increase vergence, while failed hunts do not. This supports the view that the agent represents the likelihood of success and invests in the costly vergence increase only when the hunt is likely to succeed. Vergence angle increases once prey is detected and decreases after successful eating events (Fig. 4d). Thus, agents reliably alter their vergence angles in response to the presence of prey, indicating the importance of vergence and binocularity for prey capture.

### 3.4 Virtual Experiments Reveal Agent Adaptation to Ecological and Behavioral Parameters

Beyond reproducing naturalistic hunting motifs, the framework allows systematic manipulations of ecological and sensory variables that are challenging or expensive to achieve in vivo. These "virtual experiments" test how prey statistics and sensory constraints causally shape hunting outcomes. We measured the response of our agent to five manipulations: First, as prey speed increased, capture rates declined (Fig. 5a). Second, increasing prey density led to a higher number of capture events and shorter average hunt durations (Fig. 5b). Third, vergence angle limits played a key role in the appearance of stereotyped eye vergence sequences seen in real zebrafish larvae hunting. We measured this using the mean vergence. Constraining either max or min vergence leads to a reduction in the mean vergence difference, as expected, and a mild decrease in eating efficacy (Fig. 5c-d).

### 3.5 Understanding the contribution of constraints and rewards

While the baseline configuration yields zebrafish-like hunting, we verified that each constraint both matters and is reasonably scaled by sweeping one factor at a time around the baseline (Fig. 6). These sweeps revealed that the reward terms act as effective "knobs" for shaping solution diversity: too little cost produces energetically extravagant but less naturalistic policies, whereas excessive cost suppresses pursuit structure (Fig. 6a-c). Variation across seeds further highlighted that stronger performance is consistently associated with more naturalistic vergence dynamics, suggesting that fitness-based selection favors the same stereotyped motifs observed in larval zebrafish (Fig. 6d-e). In this way, the framework provides a normative account: stereotyped hunting emerges not by chance, but as the optimal balance of sensory benefit and energetic expense under ecological constraints.

## 4 Related Work

Prior computational accounts, including bounded integrator models (Bahl & Engert, 2020) and probabilistic inference frameworks (Bolton et al., 2019), capture aspects of zebrafish hunting, but do not explain why stereotyped trajectories emerge as optimal strategies. Task-optimized neural agents have been used to model biological behaviors such as temperature-guided navigation in zebrafish (Haesemeyer et al., 2019) and electrosensory navigation in weakly electric fish (Johnson-Yu et al., 2024), and embodied neuromechanical simulations in larval zebrafish combine body physics with

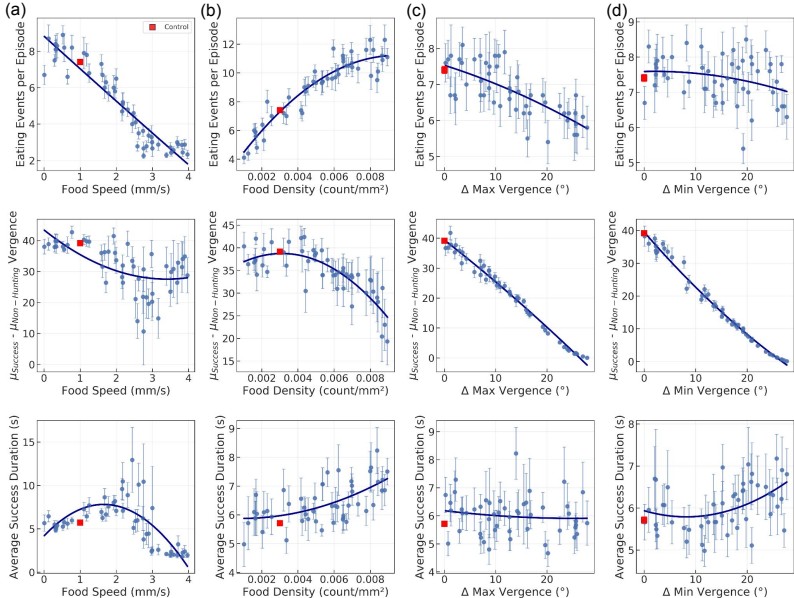

Figure 5: *Virtual experiments reveal how ecological and sensory constraints shape hunting performance.* **(a)** Increasing prey speed reduces the number of successful strikes and increases abort rates, demonstrating sensitivity to prey motion statistics. **(b)** Increasing prey density leads to more eating events and shorter hunt durations, consistent with expectations from in vivo assays. **(c-d)** Limiting max/min vergence angle reduces binocular coverage and modestly decreases hunt success in the current model. Error bars denote SEM across $n = 50$ evaluation arenas $\times$ 200 trials each. Together these sweeps illustrate how the framework can systematically vary ecological and energetic parameters to test hypotheses about zebrafish hunting, generating falsifiable predictions for future experiments.

circuit hypotheses to test constraints via systematic manipulations (Liu et al., 2024). In parallel, pursuit–evasion setups are longstanding benchmarks in reinforcement learning and multi-agent RL (Isaacs, 1999; Lowe et al., 2017; Zhang et al., 2019), but these environments emphasize adversarial dynamics and performance optimization; although larvae can rapidly learn to recognize threatening agents (Zocchi et al., 2025), we model prey as stochastic walkers to isolate how ecological and energetic constraints alone drive the emergence of stereotyped hunting strategies. Our framework differs by embedding agents in a bout-based action space with binocular sensing and explicit energetic costs, and by focusing on how ecological and energetic constraints drive the emergence of stereotyped hunting strategies, using larval zebrafish as a test case (see also Appendix 6.1).

## 5 DISCUSSION

In this work we show that a minimal recurrent agent, trained with deep reinforcement learning in a zebrafish-inspired environment, develops hunting behaviors that are both structured and stereotyped (without imitation learning or detailed biophysics). The agent's bout repertoire organizes into distinct modes resembling zebrafish strikes and adjustments (Fig. 2), successful and failed hunts diverge in trajectories and durations (Fig. 3), and vergence dynamics closely mirror in vivo recordings (Fig. 4). Systematic parameter sweeps then revealed which ecological and energetic constraints were sufficient to support these behaviors (Fig. 5), establishing the framework as a virtual laboratory for testing hypotheses about hunting.

These findings provide a normative account of zebrafish hunting as the optimal balance between energetic cost and sensory benefit. Vergence is energetically costly yet favored during hunts because it improves prey localization (Fig. 4). Forward and turn speed distributions reveal a similar trade-off: agents sacrifice forward speed in favor of faster turn speeds when hunting in order to prioritize proper

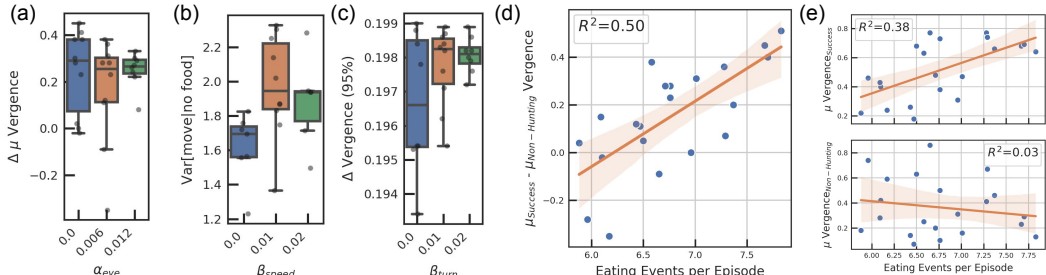

Figure 6: **Parameter sweeps reveal how energetic and sensory constraints shape hunting performance.** We train multiple seeds for each parameter configuration, noting how parameters shape solution degeneracy and naturalistic behaviors. **(a)** Effect of vergence cost $\alpha_{\text{eye}}$ on the change in average vergence ($\mu_{vergence}$) difference between hunting and non-hunting periods, normalized by maximal possible change in vergence. We find that increasing $\alpha_{eye}$ constrains the solution space to an increasingly large positive $\Delta\mu_{vergence}$. **(b)** Effect of large-move penalty ($\beta_{speed}$) on forward movement variability during exploration (when no food present). Increasing $\beta_{speed}$ requires the agent to selectively move fast only when needed, shown by the greater variance in forward speed. **(c)** Effect of large-turn penalty ($\beta_{turn}$) on vergence change across bouts. Increasing $\beta_{turn}$ forces the agent to use eyes more than turning body and reduces solution degeneracy. **(d)** For the baseline condition, difference in average vergence ($\Delta\mu_{vergence}$) across seeds scales positively with the number of eating events per episode, showing a strong correlation between hunting performance and naturalistic behavior. **(e)** Splitting out the trend in eating events per episode seen in (d) into average vergence across success periods (above) and non-tracking periods (below). We see that rewarding for eating success has led to the emergence of a tendency to be converged during Hunting and a tendency to be diverged during Non-hunting periods. For (a-c), each condition uses 8 seeds, while (d-e) use 21 seeds for just the baseline condition.

alignment and orientation to prey (Fig. 2, Fig. 3c). Together, these motifs show how ecological and energetic constraints shape stereotyped hunting strategies.

Beyond zebrafish hunting, this work illustrates how ecological and energetic constraints in DRL agents with recurrent neural networks can be turned into tools for discovery. By systematically manipulating these constraints, the virtual lab narrows the experimental space for neuroscience, pointing directly to falsifiable predictions about vergence, bout sequencing, and abort behavior. The same methodology highlights for NeuroAI how simple inductive biases—coupled action spaces, modest metabolic costs, binocular sensing—scaffold structured behavior, echoing how architectural choices shape learned representations in artificial agents. More broadly, for AI and robotics, the results suggest that constraints such as energy costs and noisy sensing are not obstacles to be abstracted away, but productive forces that can be leveraged to elicit robust, adaptive strategies.

*Limitations of the current model point toward future work.* Our agents omit biomechanics, multi-sensory cues, and detailed circuit connectivity, all of which may further constrain hunting in vivo. Behavioral sweeps show variability across seeds (Fig. 5), and in animal behavior such variability is often adaptive, supporting robustness and survival.Similar principles hold at other scales: variability helps stabilize neural systems through compensation and homeostasis (Marder & Goaillard, 2006), and in NeuroAI, has been framed as desirable solution degeneracy (Huang et al., 2025). Rather than aiming for a single converged policy, our goal is instead to reproduce the dispersion of strategies observed in nature. For zebrafish hunting, the extent of individual variability is not yet fully quantified, but ongoing ethological and neural recordings will soon allow direct comparisons between the spread of agent solutions and the diversity across animals in this task. At the same time, these caveats open clear next steps: extending the virtual lab to richer ecological contexts—including social interactions and predator–prey competition—will allow us to test how additional pressures shape stereotyped strategies. Deeper analysis of network dynamics could clarify how agent states link across levels, from neural activity to behavioral motifs to ecological constraints. More broadly, applying this sweep-based methodology to other natural behaviors will test whether the same inductive biases generalize across species and tasks, moving toward a principled account of adaptive control.

## LLM USE STATEMENT

We made limited use of Large Language Models (LLMs) to support literature exploration and the drafting of routine code such as plotting scripts. Any outputs from these tools were subsequently checked and validated by at least one author, ensuring that all code and text ultimately included in the paper were accurate and author-verified.

## REPRODUCIBILITY STATEMENT

All simulation and training code, including the zebrafish-inspired environment, agent implementation, and analysis scripts, will be released publicly upon publication, along with detailed documentation and instructions for installation and use. Hyperparameters, reward terms, and environment constants are reported in Appendix 6.4 and Tables 4. While variability in reinforcement learning studies is inevitible, we trained multiple random seeds for all parameter configurations and reported variability across seeds (e.g., Fig. 6), allowing assessment of robustness.

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

# 6 APPENDIX

## 6.1 DETAILED RELATED WORKS

**Biological foundations of zebrafish hunting.** Larval zebrafish hunting has been extensively characterized at both the behavioral and circuit levels. Behavioral studies have documented the bout-based structure of hunting sequences, vergence-linked entry into a hunting mode, systematic prey-angle halving across pursuit bouts, tightly chained pursuit with characteristic abort timing, and stereotyped strike or abort outcomes (Bianco et al., 2011; Johnson et al., 2020; Bolton et al., 2019; Zhu & Goodhill, 2023; Mearns et al., 2020). Virtual-reality assays have shown that convergent eye movements and orienting turns can be reliably evoked by visual stimuli (Bianco et al., 2011). At the circuit level, a dedicated prey-detection pathway has been identified (Semmelhack et al., 2014), and recent work demonstrates binocular integration of prey stimuli across visual areas during hunting (Tian et al., 2025). Together, these studies establish a rich empirical foundation but do not provide a normative explanation for why these motifs arise as optimal strategies under ecological and energetic constraints.

**Computational models of hunting.** Several computational models have been proposed to formalize aspects of zebrafish prey capture. One line of work emphasizes sensory filtering and accumulation: visual cues are selectively gated through habituation (Lamiré et al., 2023; Randlett et al., 2019), then integrated over time in a manner consistent with a bounded integrator model (Bahl & Engert, 2020). This account explains how prey motion evidence may accumulate to a threshold that triggers pursuit. A complementary perspective formalizes hunting as probabilistic inference: prey trajectories are represented as stochastic processes, with zebrafish effectively implementing a recursive probabilistic algorithm to predict prey positions (Bolton et al., 2019). While these models capture perceptual and decision-making elements, they do not explain why vergence shifts, prey-angle halving, or abort decisions emerge as *optimal* strategies under ecological and energetic constraints.

**Task-optimized neural agents.** Artificial neural networks trained on behavioral tasks provide complementary normative models. For example, recurrent networks have been shown to reproduce stereotyped biological strategies in temperature-guided navigation in zebrafish (Haesemeyer et al., 2019) and electrosensory navigation in weakly electric fish (Johnson-Yu et al., 2024). More broadly, training recurrent agents with reinforcement learning has been used to study emergent planning, memory, and structured representations across tasks in neuroscience and AI (Singh, 2021; Huang et al.; 2025; Simmons-Edler et al., 2025; Singh et al., 2023; Keller et al., 2025), and related embodied neuromechanical simulations in larval zebrafish combine body physics and circuit hypotheses to test how constraints shape behavior (Liu et al., 2024). These studies demonstrate how task optimization in biologically inspired settings can reveal candidate algorithms, motivating our use of this approach to study zebrafish hunting.

**Pursuit–evasion agents in AI.** Predator–prey and pursuit–evasion setups have long served as benchmarks in reinforcement learning and multi-agent RL, ranging from differential game formulations to modern deep RL implementations (Isaacs, 1999; Lowe et al., 2017; Zhang et al., 2019). These works demonstrate the emergence of pursuit strategies but typically optimize for task performance in adversarial settings, rather than probing why structured behaviors emerge. By contrast, our prey agents are simple stochastic walkers, and our focus is on how ecological and energetic constraints drive the emergence of stereotyped hunting motifs; larvae can in fact learn to recognize threatening agents in vivo (Zocchi et al., 2025), but we deliberately remove adversarial dynamics and social/multi-agent context, to isolate constraint-driven structure. Our framework therefore complements this literature but addresses a distinct set of scientific questions centered on biological plausibility and normative explanations.

## 6.2 ADDITIONAL DETAILS: AGENT DESIGN AND TRAINING

**Eye Separation and Blind Spot:** In biological larval zebrafish, the blind spot is such that there is a 1.37 mm distance from midpoint of the eyes to the start of the binocular zone in front of the eyes when eyes are diverged ((Bianco et al., 2011)). Because we model the agent as having a strike radius at which they automatically strike with some probability of success, we define the size of the

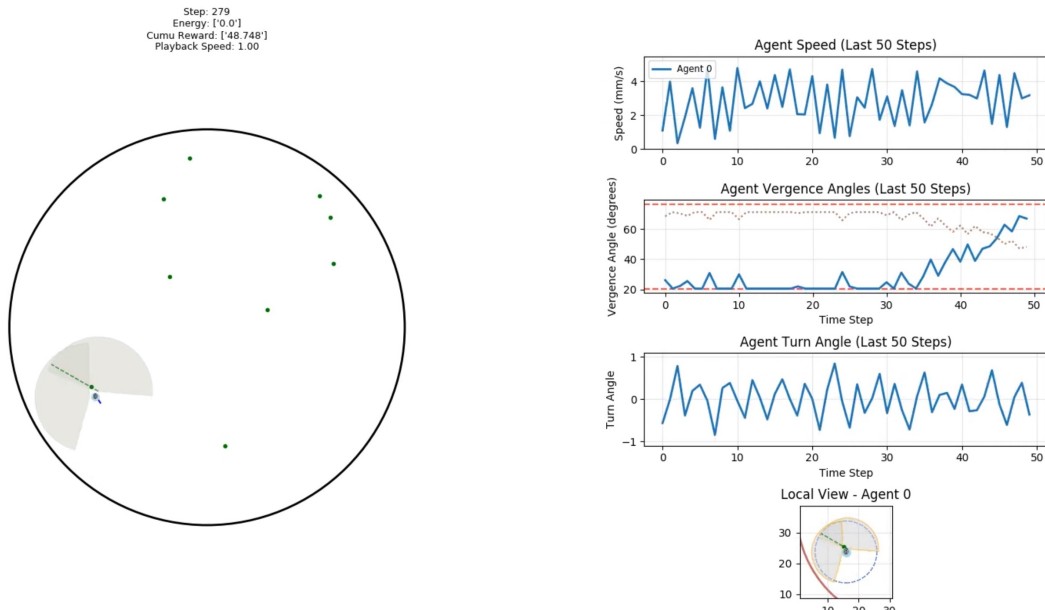

Figure 7: **Screenshot of Simulated Environment:** (**Left**) Arena snapshot showing the agent (blue), prey items (green dots), and the agent's monocular sensing fields (grey) and binocular sensing field (shaded grey wedge). (**Right**) Time series of behavioral variables over the last 50 time-steps (or bouts): forward speed (top), vergence angle (middle), and turn angle (bottom). The small inset at bottom shows the agent's local view of prey positions. High eye convergence is observed during the hunting mode.

blind spot such that there is a 1.37 mm gap between the strike radius of the agent and the start of the binocular zone. To achieve this necessary size of the blind spot, we model the eye separation as being 2 mm instead of the 0.5 mm in biological zebrafish ((Bianco et al., 2011)).

## 6.3 ADDITIONAL DETAIL: UNSUPERVISED CLUSTERING

**Clustering of movement trajectories.** PCA and K-means clustering was performed on 16-dimensional "movement trajectory" vectors, with each time point represented by a single vector. Each vector at time $t$ consists of the angular velocity and linear speed at time steps from $t$ to $t + 7$: 2 observations over 8 time points yields a total vector dimension of 16 (see Fig. 8). The number of clusters was determined by computing the silhouette score for cluster sizes between 3 and 9 and selecting the cluster size yielding the largest silhouette score (Fig. 9).

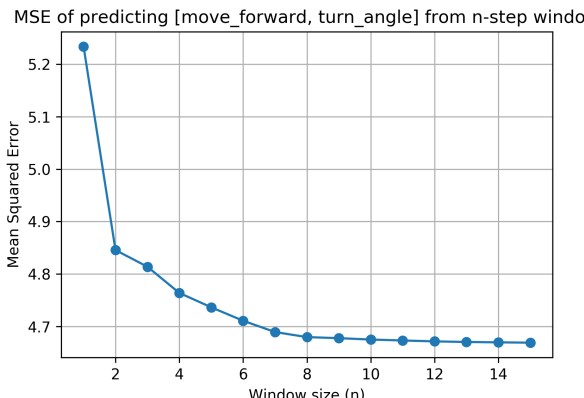

Figure 8: **Window Length for Trajectory Analysis:** Window length was determined by obtaining a characteristic timescale for movements by plotting mean squared error of next-time-step movement prediction from the previous $n$ steps. 8 time steps was selected for the time scale as considering longer time windows yields diminishing returns.

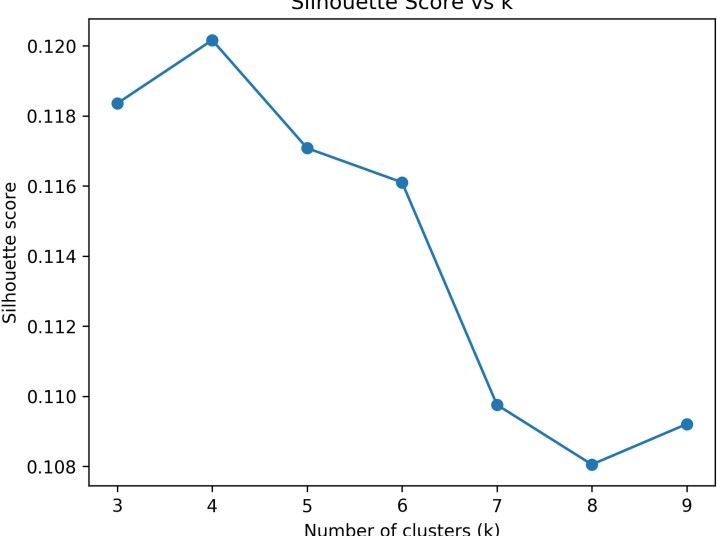

Figure 9: **Silhouette Scores for K-Means Clustering:** Silhouette scores were computed to select the optimal number of clusters for behavioral segmentation. A maximum silhouette score occurs when $k = 4$.

## 6.4 (HYPER)PARAMETER SUMMARY

| Parameter | Value | Unit | Description / Notes |
|---|---|---|---|
| max_speed | 5 | mm/s | Max speed of larval zebrafish when foraging (approx from (Fiaz et al., 2012)) |
| max_turn_speed | 7 | rad/s | Max turning speed of larval zebrafish when foraging (approx from (Voesenek et al., 2019)) |
| max_eye_turn_speed | 0.8 | rad/s | Max eye turning speed (approx from (Leyden et al., 2021)) |
| perception_field | $163 \cdot \pi/180$ | rad | Monocular field of view (Bianco et al., 2011) |
| max_left_vergence | $-43.3 \cdot \pi/180$ | rad | Left eye at maximum convergence (Bianco et al., 2011) |
| max_right_vergence | $43.3 \cdot \pi/180$ | rad | Right eye at maximum convergence (Bianco et al., 2011) |
| min_left_vergence | $-71.2 \cdot \pi/180$ | rad | Left eye at maximum divergence (Bianco et al., 2011) |
| min_right_vergence | $71.2 \cdot \pi/180$ | rad | Right eye at maximum divergence (Bianco et al., 2011) |
| bout_length | 0.125 | s | Duration of a bout (Bolton et al., 2019; Johnson et al., 2020) |
| eye_separation | 2 | mm | Distance between eyes (approx from (Bianco et al., 2011), see 6.2) |
| eye_forward_offset | 0.5 | mm | Forward offset of the eyes from the center of the agent (approx from (Bianco et al., 2011)) |
| detection_range | 10 | mm | Max food/wall detection range (at noisy, lowest resolution) (approx from (Zhu & Goodhill, 2023)) |
| eating_distribution_decay | 5 | – | Laplace decay for strike probability vs. orientation (fit to (Johnson et al., 2020)) |
| eating_angle | $80 \cdot \pi/180$ | rad | Cutoff half-angle for strike probability (fit to (Johnson et al., 2020)) |
| strike_radius | 1 | mm | Strike distance (Khan et al., 2023) |
| distance_noise_std | 0.01 | – | Std. of uniform multiplicative noise per sensor |
| penalize_move_threshold | 1.5 | mm/s | Threshold for ReLU-like penalty on forward speed |

Table 1: Simulation parameters

Table 2: Prey-related environment parameters.

| Parameter | Value | Unit | Description / Notes |
|---|---|---|---|
| food_speed | 1 | mm/s | Speed of paramecia |
| food_turn_std | $10 \cdot \pi/180$ | rad/s | Std. of (uniform) turn per step of paramecia |
| food_density | 0.003 | count/mm$^2$ | Density of paramecia in arena |

Table 3: Reward parameters.

| Parameter | Value |
|---|---|
| $R_{\text{capture}}$ | 10 |
| $\alpha_{\text{eye}}$ | 0.006 |
| $\beta_{\text{speed}}$ | 0.01 |
| $\beta_{\text{turn}}$ | 0.01 |
| $R_{\text{shape}}$ | $\sim 0.001$ |

| Parameter description | Value/Range |
|---|---|
| Environment frame rate | 8 FPS |
| RNN hidden layer width | 256 units |
| Feedforward hidden layer width(s) | 256 units |
| Model nonlinearity | tanh |
| Model layer initializations | Normal |
| RNN training steps | 4M |
| Rollout length | 600 |
| Learning Rate | $5 \times 10^{-4}$ |
| Proximal Policy Optimization (PPO) Entropy Coefficient | 0.01 |
| PPO Value Loss Coefficient | 1.0 |
| PPO Epochs | 15 |
| PPO Gamma | 0.99 |
| PPO maximum gradient norm | 0.5 |
| Generalized Advantage Estimation (GAE) Lambda | 0.95 |
| GAE steps | 2048 |

Table 4: Parameters for RL training

