# OpenReview forum: "Dissecting Larval Zebrafish Hunting Behavior using Deep Reinforcement Learning trained RNNs"
_ICLR.cc/2026/Conference — Submitted to ICLR 2026_

### Official Review · Reviewer_UPE6 · 2025-10-29

**Soundness:** 2
**Presentation:** 3
**Contribution:** 1
**Rating:** 2
**Confidence:** 3

**Summary:**

This paper investigates how sensory and energetic constraints shape hunting behavior in larval zebrafish using deep reinforcement learning. The authors develop a 2D simulator with biologically inspired agents trained via PPO with recurrent neural networks. The agents reproduce known behavioral motifs (vergence-linked tracking, prey-angle halving, abort/strike sequences) and exhibit zebrafish-like pursuit dynamics. Parameter sweeps explore how binocular sensing, bout coupling, and energetic costs jointly enable the emergence of these behaviors. The authors claim this yields a normative explanation for zebrafish hunting as an optimal energy-information trade-off.

**Strengths:**

- The paper is technically good and presents a well-described simulator with clear and replicable parameters derived from empirical data
- The authors perform a comprehensive behavioral analysis, capturing structural behavioral motifs, vergence tracking, and pursuit statistics
- A systematic constraint sweep and sensitivity analysis are conducted, exploring how various energetic and sensory parameters shape behavioral outcomes

**Weaknesses:**

- The core behavioral patterns that emerge (e.g., prey-angle halving, pursuit-abort structure) were already captured by Bolton et al. (2019) in a 3D virtual environment. This work largely re-demonstrates those results using DRL, without uncovering new mechanisms or theoretical insights.
- Inductive biases confound the "normative" claim:
  - Vergence advantage is built-in: In the default configuration, binocular sectors provide distance while monocular sectors only signal detection and include added angular noise. This design inherently favors increased vergence to access range information.
  - Strike geometry is related to biology: The success function versus angular misalignment follows a Laplace distribution fitted to larval data, effectively embedding the optimal prey-approach geometry that the agent later discovers.
   - Triangular action space: The coupling between forward and turn speeds constrains feasible pursuit trajectories, encouraging stereotyped sequences independent of learned policy optimization.
- The paper has some overclaims. For example, the paper states that the virtual lab "narrows the experimental search space and generates falsifiable predictions about behavior and neural coding". However, no explicit falsifiable predictions or neural-level analyses are presented. Claims of a "normative" framework remain qualitative and would benefit from clearer formalization.
- The relationship between energetic cost and behavioral modulation is not systematically analyzed.
- The study’s "ecological manipulations" are restricted to prey speed and density. More realistic scenarios, such as co-adaptive prey with simple evasive policies, would meaningfully test the generality of the model and could reshape strike/abort statistics or vergence dynamics.
- While the biological fidelity looks okay, the relevance for a broader AI or robotics audience is unclear. The insights are tightly coupled to zebrafish-specific behavior and may not generalize to other sensorimotor systems or artificial agents. The authors should clarify what general computational or algorithmic principles emerge.

**Questions:**

1. How are episodes initialized (agent position, prey distribution, arena size)? What determines episode length and termination? Is the reward function discrete (capture vs. abort) or continuous (distance-based shaping)?
2. Figure 2(c) shows the PCA and clustering, but the four clusters do not appear clearly separable. The trajectories plotted for each cluster also do not seem to correspond to interpretable behavioral motifs (e.g. left/right turns, straight strikes). How were trajectories encoded for the PCA, are these full trajectories or segments? The latent space does not appear to reveal structured or stereotyped movements.
3. If monocular sectors also return (noisy) distance information, does biological zebrafish behavior emerge with similar effect sizes? In order to quantify a sensory-energy trade-off, there is a need to compute Pareto front between energy expenditure and sensory accuracy
4. How does varying the fitted Laplace radius for strike success affect the emergence of ~50% angle halving and pursuit dynamics? Does modifying this shape alter approach strategies?
5. Can the authors include learning curves (reward, success rate) and performance across seeds. How many episodes were required for convergence? Comparing to random or untrained agents would contextualize the emergence of behavioral structure.
6. Would a feedforward policy or transformer-based controller (without explicit recurrence) reproduce similar behavior? This would clarify whether recurrence is essential for capturing sequential structure
7. Does the RNN encode prey azimuth or distance during the hunt? Analyzing latent dynamics (e.g. PCA or linear probes) could connect behavior to internal representations and advance the mechanistic understanding

---

> ### Author Response · Authors · 2025-11-19
>
> Thank you for the insightful feedback! We have decided to withdraw the paper to incorporate your recommendations more thoroughly and will resubmit a stronger version later.

---

### Official Review · Reviewer_avPa · 2025-10-30

**Soundness:** 3
**Presentation:** 2
**Contribution:** 2
**Rating:** 4
**Confidence:** 3

**Summary:**

This work investigated what minimal ecological and energetic constraints are sufficient for larval zebrafish, and answers it by training recurrent policies with deep reinforcement learning in a bout-based simulator that allows systematic manipulation of sensing, kinematics, costs, and environment. Despite the model’s simplicity, agents reproduce hallmark motifs (vergence-linked pursuit, speed modulation, and stereotyped approach trajectories) and quantitatively halve prey angle before strike, matching measurements. Virtual experiments and parameter sweeps show how binocular sensing, the coupling of forward speed and turning in bout kinematics, and modest energetic costs shape pursuit dynamics, strike success, and abort rates, yielding falsifiable predictions and a normative energy information account of hunting.

**Strengths:**

The paper’s strengths are clear and concrete: it introduces a biologically inspired framework in which virtual zebrafish perceive, move, and hunt through discrete bouts, enabling systematic manipulation of ecological, sensory, and energetic constraints; it trains recurrent agents with deep reinforcement learning and shows they spontaneously develop naturalistic hunting strategies without imitation learning from real zebrafish; through detailed behavioral analyses and parameter sweeps, it identifies a compact set of constraints (binocular sensing, bout kinematics, and energetic costs) minimal yet sufficient for zebrafish-like behavior to emerge; and it establishes a virtual lab that offers falsifiable predictions for in vivo experiments.

**Weaknesses:**

First, the paper does not yet show why RL is necessary: and there is no clear analysis of which motifs cannot be produced by simple rules nor evidence from RL-specific mechanisms. Second, validation is mostly qualitative; the paper should quantify agreement with real fish at the distribution level for bout-wise angle change, distance change, and vergence time courses. Third, generalization is not demonstrated; out-of-distribution tests are needed to define the applicable range of the virtual lab and to show whether the learned policy/representation transfers.

**Questions:**

1. The paper shows interesting RL agents reproducing zebrafish-like behavior under biological constraints, but several results (e.g., switching at the detection radius) look like trivial threshold effects. Even without a hand-coded baseline, please clearly state which reported motifs (e.g., angle halving, vergence dynamics, sequence rules) cannot be explained by simple thresholds or rule-based controllers, and focus the contribution on those non-trivial parts that truly need RL.
2. To argue why RL is necessary, please analyze what RL-specific mechanisms produce the non-trivial behaviors: e.g., stochastic policy, multi-objective trade-offs, or abstraction in hidden states. As this is a learning-representation venue, the paper needs quantitative analyses of internal representations (linear probes, information measures, separability across phases) to show that meaningful representations are learned.
3. Validation relies on qualitative agreement with prior empirical findings, but the level of agreement is unclear. Is it possible to provide distribution-level comparisons with real fish data that can be shown in figures for bout-wise angle change, distance change, and vergence time courses?
4. The paper does not show generalization ability. Is it possible to test out-of-distribution conditions and report how performance and behavior change? This will clarify the applicable range of  the virtual lab predictions and whether the learned policy/representation transfers.
5. In Fig. 2c the clusters do not look clearly separated (PC1/PC2 explain only part of variance), and the trajectories in Fig. 2e look very similar. Also the caption says “five clusters,” but the figure shows four.
6. The placements of Figs. 2 and 3 appear too early relative to the corresponding explanations. For readability, please move these figures closer to the main text sections where the methods and interpretations are presented.

---

> ### Author Response · Authors · 2025-11-19
>
> Thank you for the insightful feedback! We have decided to withdraw the paper to incorporate your recommendations more thoroughly and will resubmit a stronger version later.

---

### Official Review · Reviewer_H6nx · 2025-10-31

**Soundness:** 2
**Presentation:** 2
**Contribution:** 2
**Rating:** 4
**Confidence:** 3

**Summary:**

The paper develops a minimal DRL-based agent that reproduces key behavioral hallmarks of larval zebrafish hunting. The model enables virtual manipulations (e.g. changing energetic or sensory constraints) that are difficult to perform in vivo and aims to provide falsifiable hypotheses for experimentalists. The methodology is clean. I feel this paper is a good start, it needs either clarification or comparison with other baseline to pass the threshold for publication.

**Strengths:**

1. This minimal framework strengthens the authors’ claim that a small set of constraints may be sufficient to drive structured behavior, making the model a clean testbed for hypothesis generation.

2. The action space, sensory model, and timing are well-matched to real zebrafish constraints (e.g. discrete bouts, monocular vision, vergence angles), showing care in grounding the model in known biological parameters.

**Weaknesses:**

1. The agent shows plausible behavior, but most matches to larval zebrafish are qualitative (e.g. vergence increases during pursuit, prey angle halves per bout). There is no comprehensive quantitative comparison to empirical zebrafish trajectory datasets (e.g. bout frequency, success/abort statistics, distribution of tracking durations), even though such data exist (e.g. Bolton et al. 2019; Johnson et al. 2020; Mearns et al. 2020).

2. The model assumes explicit energetic penalties for forward speed, turning, and vergence. These costs shape behavior substantially. The cost parameters are selected for plausible behavior, not fitted from biological measurements of animal behaviors.

3. Although the paper frames its contributions in neuroscience terms, it does not analyze the RNN’s internal dynamics to make contact with known zebrafish circuits (e.g. the optic tectum, hindbrain saccade areas).

**Questions:**

1. Other recent studies (e.g. Harpaz et al. 2021, Lagogiannis et al. 2020, Zocchi et al. 2025) include social or adversarial dynamics. The current model uses a single agent hunting random-walking prey. Can the authors discuss how this frame work may extend to adaptive prey or multi-agent scenarios?

2. The key claim is that zebrafish-like behavior emerges as the optimal balance between energetic cost and sensory benefit. But the energy cost is manually impose, the agent was implicitly guided toward efficient behavior. Can you reframe it as hypothesis testing such that if energy cost is causal (instead of other constraints that did not get tested here, like accuracy-only or uncertainty minimization)

---

> ### Author Response · Authors · 2025-11-19
>
> Thank you for the insightful feedback! We have decided to withdraw the paper to incorporate your recommendations more thoroughly and will resubmit a stronger version later.

---

### Official Review · Reviewer_xekH · 2025-11-01

**Soundness:** 1
**Presentation:** 2
**Contribution:** 1
**Rating:** 2
**Confidence:** 3

**Summary:**

This paper utilizes Deep Reinforcement Learning to develop hunting strategies similar to those of zebrafish by applying empirical biological constraints and designing a reward function without explicitly learning from empirical data. The behavior of the trained agents aligns qualitatively with real zebrafish behavior, such as eye vergence and steering patterns. Additionally, with the trained agent, the authors investigate how altering ecological variables affects hunting efficacy.

**Strengths:**

The authors qualitatively replicated the bout behaviors of zebrafish with a simple but effective reward function while following the biological constraints of the species based on existing literature. The research problem tackled in the paper is interesting to multiple disciplines, including neuroscience and deep learning. The illustrations and plots in the paper are easy to follow.

**Weaknesses:**

The contribution of the paper is not significant enough for ICLR because (1) the conclusion of the paper (providing a normative account for why pattern emerges) is weak, (2) the similarity between trained agents and real zebrafish behavior is not thoroughly proven, (3) the signficance of the work on future neuroscience or AI research is unclear, and (4) the technical aspects behind the deep RL models are too simple for ICLR.

(1) Underwhelming conclusion

Most of the findings in the paper are trivial and do not warrant the use of a deep RL model. For example, the effect of prey density and prey speed on hunting success in Section 3.4 is trivial. The conclusion that the proposed framework explains why stereotyped hunting emerges under ecological constraints is underwhelming, as it should be a well-known fact in the biology community.

(2) Weak quantitative evidence behind hunting pattern similarity

The authors only selected a few general hunting patterns, such as halving of prey angle [Bolton et al., 2019], as evidence for the naturalistic behavior of the trained agents. Stricter quantitative analysis between agent behavior and real zebrafish behavior is necessary. For example, the authors should capture real zebrafish motion and compare it with the agents' motion to prove that the agents can learn naturalistic hunting behavior. As there is no real fish data in any of the plots, the evidence shown is only qualitative at best.

(3) Unclear application

While the authors mentioned that the work can be useful for neuroscience and robotics research, it is unclear how the framework can be applied to those fields. The claim that "energy costs and noisy sensing are ... productive forces that can be leveraged to elicit robust, adaptive strategies" is already well known in the RL community. Again, this weakness is closely related to the (1) weak conclusion of the paper in general.

(4) weak technical contribution

The deep learning approach used in the paper is too simple for ICLR.

**Questions:**

What is the most significant contribution of the work to the deep learning community? Why is it significant and novel to the community?
Is it technically challenging to replicate the hunting behavior of zebrafish?
How rigorously have you proven the behavioral accuracy of the trained agents?

---

> ### Author Response · Authors · 2025-11-19
>
> Thank you for the insightful feedback! We have decided to withdraw the paper to incorporate your recommendations more thoroughly and will resubmit a stronger version later.

---

### Author Response · Authors · 2025-11-19

We thank all reviewers for their thoughtful and constructive feedback. Your comments highlighted important directions for further experiments and analyses that will significantly strengthen the paper. Because completing these additions goes beyond what is feasible during the rebuttal period, we have chosen to withdraw the submission and undertake a more substantial revision. We look forward to resubmitting a more complete version that fully addresses your insights. Thank you again for the time and care you devoted to our work.

---

### Meta-Review · Area_Chair_3ev5 · 2025-12-29

**Summary:**

All four reviews of this submission lean toward rejection. Reviewers raised several major concerns about the submission’s technical method and experimental results: 1) Most reviewers shared concerns with the lack of a comprehensive quantitative analysis of the hunting behaviors reported in the experiments; 2) Several reviewers questioned the technical novelty and contributions of the method, especially the necessity of applying RL; 3) The generalizability of the proposed method needed more validation, and its application in AI/science in its current form remained unclear to some reviewers.

**Reviewer Concerns:**

The authors did not submit any rebuttal to address these concerns.

**Reviewer Scores:**

No changes because of the lack of a rebuttal.

---

### Decision · Program_Chairs · 2026-01-26

Reject